# A Multilingual App for Providing Information to SARS-CoV-2 Vaccination Candidates with Limited Language Proficiency: Development and Pilot

**DOI:** 10.3390/vaccines10030360

**Published:** 2022-02-25

**Authors:** Eva Maria Noack, Jennifer Schäning, Frank Müller

**Affiliations:** Department of General Practice, University Medical Center Göttingen, 37073 Göttingen, Germany; jennifer.schaening@med.uni-goettingen.de (J.S.); frank.mueller@med.uni-goettingen.de (F.M.)

**Keywords:** language barriers, limited language proficiency, migrants, health equity, vaccine hesitancy, SARS-CoV-2 pandemic, COVID-19 vaccination, sign language

## Abstract

Language barriers are obstacles in receiving vaccinations against COVID-19. They jeopardize informed consent, vaccination safety, and a positive immunization experience. We have developed a multilingual app to overcome language barriers when dealing with vaccination candidates with a limited proficiency in the locally spoken language. We applied the Spiral Technology Action Research (STAR) model to create the app within a discursive process involving healthcare professionals (HCPs) from vaccination sites, literature searches and guidelines, and field trials at vaccination centers. In a real-world pilot test, we assessed the usability and feedback for further improvement. Our efforts resulted in an app that facilitates communication with vaccination candidates in 40 languages, each with over 500 phrases that can be played back or displayed as text. In the pilot test, the app demonstrated its usability, and was well accepted by the vaccination candidates (*n* = 20). The app was mainly used to inform about the risks and benefits of the SARS-CoV-2 vaccination. Some HCPs struggled to navigate the comprehensive content and the pilot test exposed the need for additional phrases. The STAR model proved to be flexible in adapting to dynamic pandemic conditions and changing recommendations. This multilingual app overcomes language barriers in healthcare settings, promoting vaccines to migrants with limited language proficiency.

## 1. Introduction

Increasingly, healthcare providers all over the world attend to patients with different cultural, ethnic, and linguistic backgrounds. Providing healthcare to patients with limited proficiency in the locally spoken language poses challenges to those seeking help and the attending healthcare professionals (HCPs) alike [1,2]. In the ongoing vaccination campaign to contain the COVID-19 pandemic and immunize the broadest population, reaching all people and communities regardless of their ethnic or socioeconomic background is the key to success. Language discordance between patients and healthcare providers is a major barrier to the equitable provision of health services. Systematic reviews have shown that recent migrants tend to make less use of preventive services, including immunization, than the population without a migrant background [3,4]. This has implications for the health of both the individual and the entire community. Given their average health status and social circumstances, migrants tend to have a higher risk to be exposed to and infected with SARS-CoV-2, whilst those living under precarious conditions are at an increased risk of an outbreak of COVID-19 [5,6]. Several studies suggest a higher vaccination hesitancy among ethnic minorities [7,8,9]. Articles report that a lack of appropriately worded information, alongside negative experiences within a culturally insensitive healthcare system, constituted relevant barriers to access and the uptake of the SARS-CoV-2 vaccination [10,11].

According to the doctrine of informed consent, an HCP is obliged to inform the person to be vaccinated or their legal guardian in an understandable way about the benefits and potential risks of vaccination and obtain consent. The informed consent talk should include information about the disease to be prevented and its treatment options, information about the benefits of vaccination, contraindications, the application of vaccination, the onset and duration of vaccination protection, behavior after vaccination, possible adverse drug reactions and vaccination complications, and the need for and dates of follow-up and booster vaccinations [12]. It is very challenging to obtain informed consent with all its facets from non-language proficient vaccination candidates, i.e., to adequately convey immunization information and ensure that they understand the implications; these challenges cannot be solely addressed by vaccine information booklets in different languages. As SARS-CoV-2 vaccines are all new to the market, giving respective information, obtaining a basic medical history, and answering vaccination candidates’ questions are of particular importance—especially, to enhance vaccination uptake among foreign-language speaking vaccination candidates. The situation is aggravated by the fact that Germany, unlike other countries in Northern Europe or the United States, does not regularly cover costs for interpretation services in healthcare, and thus returns the issue to vaccination candidates and their doctors.

In this respect, communication is a major challenge to achieve a high SARS-CoV-2 vaccine uptake among people with no or very limited proficiency in German.

### Objectives

To address these problems, we developed an app together with aidminutes GmbH, a German e-health service provider of web-based apps that enable HCPs to provide legally compliant information on all vaccines approved by the European Medical Agency (EMA) in 40 different languages. By providing information about vaccines and the vaccination process in each vaccination candidates’ native language, the project aims to overcome language barriers, but also any distrust and vaccine hesitancy. Additionally, HCPs can use the multilingual app to obtain vaccination-relevant medical history from vaccination candidates, obtain informed consent, and document all information communicated. The content was discussed and approved by the Robert Koch Institute (German National Institute for Public Health); the set of supported languages was decided together with the German Federal Ministry of Health.

In this article, we describe how we developed this app, the app’s content, structure, and functionality and report on our first experiences within a pilot test on migrants. We aim to highlight important considerations relating to the app’s development and respective implications. 

## 2. Materials and Methods

### 2.1. Framework of the Development Process: The STAR Steps

We aimed to create an intuitive vaccine communication app that enables HCPs to collect or provide relevant information quickly and without previous training. To achieve this aim, we applied an agile developmental approach, informed at all relevant stages by potential users and their ideas, feedback, hints, and wishes. Additionally, both conceptual aspects and usability testing of prototypes were designed to ensure that the final app met the end users’ needs.

For this purpose, we chose the Spiral Technology Action Research (STAR) model [13], a theoretical framework for developing health promotion interventions [14,15]. Known as a quality improvement approach, the Plan, Do, Study, and Act (PDSA) cycles build the model’s core components. Accordingly, we divided our app development process into the cycles: (1) Listen, (2) Plan, (3) Do, (4) Study, and (5) Act, with repetitive steps. In each step, new information allowed us to make important modifications in response to the fast-moving pandemic situation and constantly emerging knowledge about COVID-19 and SARS-CoV-2 vaccines.

(1) Listen: Listening to the target group(s) and population(s), we found out their needs and determined how they identify with technology. To make the app intuitive and most suited to the vaccination process, we interviewed medical, organizational, and administrative staff at vaccination centers who provided us with the necessary information to combine theory and practice. 

Later, in the translation stage, we listened to the community of sign language users. This discourse revealed that communication with hearing-impaired vaccination candidates was yet another linguistic challenge. As per the STAR model, we were able to integrate sign language videos displayed on a smartphone screen into the app as well (See Appendix A). 

(2) Plan: After the target groups’ needs were identified, we devised a technical and organizational implementation plan. Figure 1 shows one round of the development process. We started by considering necessary functions. Our previous experience in developing a multilingual app that facilitated conversations between paramedics and foreign-language patients during medical emergencies [16] helped us to create an intuitive, aesthetically pleasing interface for the current app [17]. After reviewing and reconsidering our previous contemplations about the app’s user experience (UX) design and user interface (UI) principles, we defined the final UX and UI modification. 

The main challenge was to compile, screen, and organize the content. To find relevant content, we (a) perused the summaries of product characteristics (SPCs) published by vaccine producers of all vaccines approved in the European Union [18,19,20]. SPCs contain the benefit–risk assessments healthcare professionals need in order to make decisions and are continually updated. (b) We reviewed safety information published by the Paul Ehrlich Institute, Germany’s Federal Institute for Vaccines and Biomedicines [21]. (c) We included recommendations, vaccine information and consent forms, and medical history questionnaires published by the German Standing Committee on Vaccination which are continually updated [22]. (d) We observed process flows at German vaccination centers to become acquainted with actual vaccination practice beyond formal procedures. These observations included the registration process (where vaccine candidates prove their identity and vaccine eligibility), the waiting area (where vaccine candidates read vaccine information and fill out medical history questionnaires), pre-vaccination consultations with doctors (that comprise taking a short medical history and informed consent talk), the vaccination itself, after-vaccination care and ‘check-out’ (handing in the papers for documentation). We commissioned translations, fully aware that this process requires feedback loops to prevent mistranslations. We also had to ensure that the target community was always involved to ensure authenticity. 

(3) Do: After scanning the material, we pre-screened and synthesized the relevant information to refine the content. The scope of an informed consent talk and information required must be personalized [12]; to allow for this variability across talks, the app’s content had to be comprehensive. On some phrases, we sought legal opinions from specialized lawyers to ensure the information given in the informed consent talk and during the medical history taking were legally binding. Then, we phrased the content in plain language, initially German, that is easy to understand, and ensured its translatability. Short, catchy sentences had to be used; convoluted sentences were rephrased, or the passive voice was changed to the active. We had to omit figurative language or phrasing that can only be understood in German. To ensure that the accuracy and comprehensibility was retained in other languages, the phrases were translated and audio-recorded by professional interpreters and then played to native speakers to check whether they were understood. In addition, the content was partly translated back into German by other interpreters as part of the feedback loop. 

“The Association for German Sign Language” supported the production of videos to include sign language, and gave advice on standards, such as the contrast and size of the screen. Finally, we set up an initial app structure for the comprehensive content.

(4) Study: We discussed the selected pre-phrased sentences with HCPs working at vaccination centers and conducted field trials on-site. The field trials included informal observations of processes, conversations with vaccination candidates, and informal talks with staff, and were done in several iterative steps. Initially, we observed processes to find out what was communicated at which step and by whom. Later, borderline cases (e.g., vaccination candidates who forgot to bring proof of vaccination or refused vaccination, vaccination not indicated for medical reasons, medical emergencies after vaccination) were defined and discussed with the staff. 

Next, we had the layout and navigation of app prototypes reviewed by HCPs. Their feedback helped us to improve navigation by refining the app structure. The study step is a continuous process, i.e., post-release monitoring continued to verify whether the app was functional. 

(5) Act: After incorporating feedback from the previous steps, the app was launched and disseminated to HCPs. Aligned with the rapidly evolving pandemic, the app development process took six weeks between conception and release. That is why it was and is important to react to continuing feedback and implement mechanisms to update the app based on the results from the ongoing process of continual learning and improvement. Post-release, we provided automatic updates bimonthly. 

### 2.2. Pilot Test

After the app’s release, we designed a real-life pilot test to determine how HCPs used the app in interactions with foreign-language speakers. The primary objective was to determine the usability of the app; the secondary objective was to gather information for continuing improvement. For this purpose, an observation protocol was created. Basic sociodemographic and consultation-related data were collected on the vaccination candidates (gender, age group, accompanying person, presence of interpreters, language spoken). The number of played phrases during registration, informed consent talks, history taking, and informed consent receipts was counted by the observer. Additionally, the thematic content of the phrases (such as “behavior after vaccination”) was coded into choice boxes. For each subarea, a global rating was given on a 5-point Likert scale to indicate whether communication with the app worked. This global rating comprised the technical aspects of the device (e.g., was the output loud and understandable enough?), the user perspective (was the HCP able to find the desired phrase or was it missing?), and the vaccination candidate’s perspective (was the vaccination candidate proactive?). A free text field was provided for each of these communicated topics, where further information about usage behavior was described. Situational peculiarities and processes were documented in additional short memos for each case.

The observation protocol was concluded by three questions on the overall assessment, which also had to be rated on a 5-point Likert scale. These questions focused on the overall co-operation/compliance of the vaccination candidate within the situation where information was provided, and questions were posed by the app and whether it was possible to communicate with the vaccination candidate in German at all.

The paper-based two-page observation protocol was scanned with the Evasys digital survey system (Evasys GmbH, Lüneburg, Germany). Data was exported to and analyzed in SPSS (IBM, Armonk, NY, USA) using descriptive univariate methods including means and standard deviations. Free-text comments and handwritten memos were transcribed and grouped thematically according to a content analysis approach.

In order to get as unbiased an impression as possible of how the app was actually used, HCPs did not receive any instructions beyond a short introduction showing the basic functions. Particularly, we did not specify how the app should be used for consent talks and left this to the decision of the HCPs. We did not set any time restrictions for the use of the app. The observers did not intervene, help, or comment during the app use situation. HCPs could use the app on their own smartphones or on one of the tablets we provided (Apple iPad mini 4).

## 3. Results

We have developed and released a multilingual app for HCPs that facilitates communication with vaccination candidates with limited proficiency in the local language (Video S1). The app enables HCPs to provide effectively legally compliant information about the risks and benefits of the SARS-CoV-2 vaccination, to obtain informed consent, and to guide vaccination candidates through the vaccination process. The app is designed to be operated by HCPs on their (work) smartphones or tablets. Thus, vaccination candidates neither need to install an app nor have a smartphone at all.

### 3.1. Content and Structure

Our observations of process flows revealed that the app needed to comprise phrases along the entire vaccination process. This included the categories: registration process, medical history taking, informed consent talk, obtaining informed consent, directions for getting around the vaccination center, and information on follow-up.

Since the potential side effects and risks of the administered vaccines differed, the content, i.e., what had to be communicated or asked, was specifically designed for each vaccine. The app, therefore, was adapted to contain relevant phrases for all SARS-CoV-2 vaccines approved in Germany. At the app’s release, these were Comirnaty (BNT162b2, BioNTech, Mainz, Germany / Pfizer, New York City, NY, USA), Spikevax (mRNA-1273, Moderna, Cambridge, MA, USA), and Vaxzevria (AZD 1222, AstraZeneca, Cambridge, UK). Content for the COVID-19 vaccine Janssen (Ad26.COV2-S, Johnson & Johnson, New Brunswick, NJ, USA) was added later after its introduction to the European market. 

Since some vaccination candidates had already been vaccinated with other non-EU-approved vaccines before coming to Germany, the app was designed to asked questions like whether the vaccination candidate had received immunization with Sputnik V (Gamaleya Research Institute) or Coronavac (Sinovac). The included phrases also considered the vaccination candidate’s gender, i.e., women could be asked about possible pregnancy.

Since vaccination candidates needing assistance brought along family members, friends, or other trusted persons, part of the informed consent talks took place with the involvement of a third party. We discovered that the app had to encompass questions and information addressed to such third parties as well. This was also imperative when vaccination candidates were not able to give informed consent themselves and the informed consent talk had to take place with relatives or legal guardians. 

Since answers to open questions were not reliably understood, the app had to work with fixed phrases spoken as closed questions, i.e., requests to do or show something (e.g., a vaccination certificate), instructions, information about the process, or an announcement of measures. 

Based on our field trials, we grouped the app’s comprehensive content into seven categories for fast retrieval: (1) registration process, (2) informed consent talk (information on the chosen vaccine), (3) medical history taking, (4) information on the second vaccination, (5) obtaining informed consent, (6) post-immunization follow-up, and (7) further information. Figure 2 shows a screenshot of the first six of the seven categories and their main functions. The order of the categories reflects the chronological sequence of steps at the participating vaccination centers. 

### 3.2. Languages, Functions, and Navigation

#### 3.2.1. Supported Languages

There are no official German statistics about the native languages of migrants, “tolerated” refugees, and asylum seekers, nor about migrants’ language use or language skills. We therefore proposed a set of languages based on the demographic statistics of migrants and the countries of origins of recently immigrated refugees and asylum seekers. As every individual living in Germany is eligible for COVID-19 vaccination regardless of nationality or insurance status, we chose to include a large number of languages in order to guarantee access to vaccination and adequate information for as many people as possible. The app currently supports 39 languages (see Table 1). Every supported language (except German sign language) can be used as both the operating language and output language. Thus, the app can be potentially deployed in various countries by any HCP familiar with one of the supported languages. To inclusively address people with hearing impairments, the app also incorporated German sign language by means of video output.

#### 3.2.2. Functions and Navigation Features

To start, the app creates a new vaccination case. Using this case structure, we were able to document app-based communication and assign the log to the vaccination candidate. Next, the HCPs select the vaccine to be administered. For safety reasons, the selected vaccine is visible also within categories. 

The vaccination candidate’s gender and preferred language are selected in a third step. Drawing on our previous experience with a multilingual app [16], we provide two different language sorting styles for this function: languages can be chosen from either a geographically grouped or alphabetically sorted list. 

Both sorting styles allow vaccination candidates to recognize and choose their preferred language. This resulting design proved helpful when an HCP struggled to identify the vaccination candidates’ spoken language. Language comprehension is checked in the following step (Figure 3). Next, a short audio introduction is provided to the vaccination candidate about why and how the tool is used.

Next, the above-mentioned categories appear, which form the core of the app (Figure 2), grouping phrases in typical conversational situations. Within a category, the needed phrases (Figure 4) are selected and played. 

Observations and discussions with staff at vaccination centers revealed that the app needed to provide all phrases as both audio and text to include vaccine candidates who could neither read nor hear (Figure 3 and Figure 4). The displayed text also ensured that confidential matters can be addressed discreetly (e.g., pregnancy). Therefore, any phrase could be chosen to be played back or shown to the conversational partner.

Phrases in sign language were provided as multi-perspective videos (upper part of the body from the front and obliquely from the side and frontal portrait, Figure 5). Based on feedback from the hearing-impaired, features like higher image contrast ratios were implemented to meet their requirements. 

To enable a change of address during the app-supported communication, e.g., to an accompanying person, the user can toggle from each navigation level, i.e., in the main categories view (Figure 2) or at the selected question (Figure 4). 

To avoid long lists of items that require attentive scrolling and therefore slow down handling, we grouped related questions within the seven main categories. Filter questions answered in the affirmative triggered further related questions, e.g., fever, allergies, pregnancy (Figure 6).

To additionally facilitate navigation, we introduced shortcuts between related phrases that allowed for direct navigation from the medical history-taking category to the obtaining informed consent category without having to return to the category view. This allows HCPs to navigate through a logical sequence of phrases typically used at vaccination centers. At the same time, the navigation is neither fixed nor binding at any point. This makes HCPs free to decide which phrase to use next, including already previously played or displayed phrases, at all times.

Phrases that are valuable in an emergency can be accessed quickly by clicking on the menu positioned center bottom. These phrases were derived from the ‘rescue.app’ paramedic tool [16,17,23].

All phrases played and vaccination candidates’ responses can be logged, and the course of the conversation can be displayed at any time. After the consultation, the log can be saved, and the data, local infrastructure permitting, can be transferred to the vaccination center’s digital documentation system (Figure 2). 

### 3.3. Pilot Test

The app was initially developed for use at vaccination centers. However, it turned out that these centers were mainly attended by German-speaking vaccination candidates. To target hard-to-reach groups more effectively and increase the vaccination uptake in these groups, mobile vaccination outreach teams were formed in the summer of 2021 which offered vaccination outside of mass vaccination sites and doctor’s practices. To assess encounters with foreign-language vaccination candidates in a real-life setting, we observed the app’s use by mobile vaccination teams in Leipzig, Saxony, Germany. 

Previous knowledge and experience with the app varied among HCPs on the mobile team: all HCPs stated that they had already heard about the app, some had already installed it on their smartphones but never used it, and some had used the app very extensively (in one case over 200 times) and included the app in their daily routine, e.g., using Bluetooth loudspeakers to support audio comprehension. 

We observed and documented the app use with 20 vaccination candidates across six outreach deployments. The age groups of vaccination candidates (12 male, 8 female) ranged from 41 to 65 years old (n = 11) and 18 to 40 years old (n = 8). The app languages used here were: Arabic, Romanian, Spanish (Latin American Spanish), Vietnamese (all 3 times); Albanian and English (both 2 times); and Thai, Polish, Slovak, and Russian (each 1 time). The vaccines used were Comirnaty (n = 16) from BioNTech/Pfizer, and Spikevax (n = 4) from Moderna. In n = 4 observed cases, the vaccination candidate brought a family member as a lay interpreter to the appointment.

An average of 10.4 (SD 7.8, range 2–27) phrases were played per case, with information about the administered vaccine conveyed in all cases. Here, the risks and side effects were mainly communicated. In 75% of the observed cases, questions about vaccination history were asked, particularly about allergies and fever. In 75% of cases, the app was used to explicitly ask for informed consent to vaccinate. Phrases that aimed to support the registration processes at vaccination centers (n = 2) were rarely used. Most vaccination candidates were asked about their handedness. Additional statistics are provided in Table A1.

The perceived compliance and understanding, e.g., whether vaccination candidates interacted with the app, was rated on a 5-point Likert scale (1 = yes, totally; 5 = no, not at all). The ratings on the different phrase categories ranged between 1.00 and 1.13. The vaccination candidates’ perceived German language skills were rated on a similar 5-point Likert scale were low (Mean 4.0, SD 1.3). No app usage was cancelled or refused by vaccination candidates. 

Categorized memos are shown in Table A2. These memos revealed that the app does not yet account for some content requested by HCPs. For example, although it was possible to convey that a second vaccination is needed for full inoculation, a function to schedule a second appointment was missing. Moreover, a phrase imposing a post-vaccination observation period of 15 or 30 min after vaccination was not included. Some functions were included on principle, e.g., information that a single vaccine shot is sufficient to reach full inoculation if the vaccination candidates had had a proven COVID-19 infection. However, the question “Did you have a proven COVID-19 infection in the last 6 months?” was missing. Likewise, the recommendation to take nonsteroidal anti-inflammatory drugs against transient fever after vaccination was included in the general information on vaccination, but one HCP wished to be able to provide this information independently and immediately after vaccination. Other requested phrases revolved around information on the digital European Vaccination Certificate, the risk of post-vaccination myocarditis, and how potentially allergenic additives in mRNA vaccines like polyethylene glycol could be better explained.

Additionally, we experienced some usability issues and flaws. Initially, some phrases could not be played. In two observed cases, the phrases needed were not found immediately or were assumed to be in another category. Navigation through the app’s multi-phrase comprehensive content often proved challenging and time-consuming. 

Some feedback focused on phrase comprehension: one vaccination candidate obviously did not understand the question about handedness. Another phrase in the app was posed in a way that it could not be answered with a yes/no gesture (“Would you prefer to be vaccinated sitting or lying down?”).

Some HCPs reported that use of the app made staff, accompanying persons, and vaccination candidates feel more relaxed about the situation, emphasizing that the app helped reduce the fear of vaccination. However, some HCPs who struggled with browsing and navigation perceived the app usage as more of a burden than a timesaver. In one isolated case, the lack of a phrase or the perception that the app was not up to date resulted in reduced trust towards the app.

## 4. Discussion

### 4.1. Discussion of the Results

Language barriers are a major factor preventing foreign-language speakers from receiving a safe and sincere vaccination experience [7,24]. Interpreting resources are often limited. We have developed a comprehensive, fixed-phrase, multilingual interpreter app that can provide relevant information and asks relevant questions for the informed uptake of SARS-CoV-2 vaccinations in 39 different languages. The app’s use is flexible and can be adapted to each vaccination candidate’s information needs and previous experience. All phrases can be played back as audio output or displayed as text in the respective languages. It also proved to be suitable for vaccination candidates with impaired hearing or vision, or who are illiterate. Additionally, German sign language is supported by video content. All information presentation and informed consent-obtaining procedures are designed to be legally compliant. The communication process can be documented and archived. The app was freely available in various app stores (aidminutes.impfen) during the 2021 vaccination campaign and has been installed over 73,000 times. To our knowledge, it is the first and only app of its kind.

In many Western countries, language barriers have to be overcome to encourage migrants and foreign-language speakers to get a SARS-CoV-2 vaccination. As a result, these and other hard-to-reach communities lag significantly behind the autochthonous population in terms of vaccination rates [7,8,9]. Funded by the German Federal Ministry of Health, the app we present here proved it could supplement governmental efforts to increase vaccination willingness in migrant populations, by improving communication on both a linguistic and cultural level. Moreover, it effectively dismantled prejudices by helping HCPs provide safe vaccine information and obtain informed consent from vaccination candidates with limited language proficiency in the national language. 

We carried out our observation of app use with a mobile vaccination team located in Leipzig, East Germany. Designed to foster workflows at mass vaccination sites, the HCPs participating in this pilot test reported that the app was barely used at regional vaccination centers. The presumed reason was that few foreign-language speaking vaccination candidates presented themselves because the hurdles in accessing these centers were too high. The vaccination candidates were required to schedule an appointment by phone, fill out forms, and read vaccine information in German in advance. Conversely, the outreach deployments of mobile vaccination teams organized according to a walk-in concept achieved better uptake for vaccination candidates with a migrant background. 

Although the app was not designed for mobile teams, its use proved effective in principle. Whilst vaccination candidates in all observed cases interacted well with HCPs using the app and accepted this form of communication, some HCPs struggled with its use. Browsing through the content to find phrases sometimes proved time-consuming, especially for first-time users. 

In principle, the authors developed an app with a sequence of fixed phrases, thus making it more structured and much more user-friendly. Theoretically, an app that can be administered by the vaccination candidates themselves, that guides them through the medical history and vaccine information phases would be most useful. However, both of these options would not have enabled the provision of information that complies with national and international regulations. These statutes dictate that informed consent talks must be ”customized” to the vaccine candidates’ need for information and their individual risk constellation in order for them to make an informed decision. 

Informed consent must be obtained by an HCP. Furthermore, vaccination candidates must be given the opportunity to ask questions. In order to include all possible communication needs that might arise during the vaccine-related informed consent talk, the app was designed to be legally and medically comprehensive with an overall total of over 500 phrases for each supported language. For example, information phrases about the active mechanism of all available vaccines with varying degrees of detail were included in the app. Therefore, the requirements of legally sound information presentation and obtaining of vaccine-related informed consent was realized. 

Nevertheless, routine vaccination procedures are carried out much more pragmatically. This was also revealed by our observations: in 25% of the observed cases, the question to consent to vaccination via the app was not asked at all. Since no other language interpretation was available, it must be assumed that consent was assumed in these cases by the HCPs without them directly obtaining consent. In our observations, we noticed long waiting times for vaccination candidates and attempts by team staff to catch up with administrative parts. This resulted in high time pressure for the mobile vaccination teams that may have affected how the app was actually used. 

The fact that that some HCPs had already been using our multilingual app effectively in their daily routines prior to our test observation suggests that regular use and exposure to this tool enhances user friendliness and eliminates the difficulties that first-time users experience. These findings are also consistent with our experience gained from our similar app for paramedics [17].

### 4.2. Limitations and Strengths

In terms of limitations, the pilot test, field trials, and observations exposed usability issues and flaws in our app: some phrases were missing or hard to find. We aim to resolve these shortcomings in upcoming updates by adding additional phrases and restructuring some categories. 

In addition to linguistic barriers, there may be further communication problems in medical encounters with ethnic minority patients. These include (cultural) differences in explanatory models of diseases, differing preferences for shaping the doctor–patient relationship [25], and a divergent perception of doctor–patient communication [26,27]. This applies not only to SARS-CoV-2 vaccinations, but also to other preventive measures that might not reach the target minority population sufficiently [26,27]. In this context, future research should focus on listening to healthcare recipients, i.e., finding out how foreign-speaking patients and minority groups receive and interpret medical information.

The main impetus behind the development of this multilingual app was to ensure rapid availability within two months of starting to support the ongoing German vaccination campaign. In parallel, development took place in a volatile situation where vaccine recommendations were constantly being changed and new vaccines were approved. Given the dynamics of the pandemic and vaccination campaigns, the applied STAR model was well suited for this flexible operation under evolving circumstances. 

As part of this process, the user perspective of vaccination center staff was taken into account, with vaccinating staff participating in the app’s development. Due to the need for pragmatic implementation, the perspective of migrant groups was not directly collected. This meant that vaccination myths and misinformation, proven to be more prevalent among certain migrant communities [28], could not be addressed specifically by the app.

The pilot test aimed at assessing a general feasibility for using our tool in a real-world setting, at identifying potential hurdles in usage, and serving as feedback for the development process. This pragmatic pilot test with 20 observed uses was, by its very nature, not suitable to assess the efficacy of the app as an intervention. Further research is needed to determine whether vaccination candidates are better informed through the app than through multilingual information forms. Nevertheless, it must be noted that multilingual forms can never replace a legally and ethically sound informed consent talk. Here, the app offers a significant improvement.

Despite challenges experienced by some users, the basic concept seemed quite successful. Nevertheless, observations can be biased. This may be especially true when actions are monitored in the presence of language barriers. Our observational study, therefore, was not able to investigate how the app was perceived by the vaccination candidates. This meant that it was ultimately not possible to assess if vaccination candidates took a skeptical or approving stance to the app. Furthermore, we did not ascertain whether all phrases were understood or if major errors existed in the translations. These aspects are thus a good starting point for further research.

### 4.3. Outlook

The plan is to continuously refine the app and adapt it according to current guidelines. In the near future, the app will offer calming sentences and explain the procedure to children by addressing them directly in a child-friendly manner. Here, we face the complex communicative setting of speaking to children and parents, providing adapted information, dispelling concerns, and dismantling prejudices.

In principle, the use of the app is not limited to Germany, as any supported language can be used as both operating language and output language. The approach could thus also support many international vaccination campaigns. At present however, only the recommendations of the German Standing Commission on Vaccination are consistently supported.

To verify whether the app can be used in the dynamic process of vaccination campaigns, further study-accompanying field application studies should be carried out. For example, more user reports and observations are needed to adapt the tool to the changing vaccination process.

## 5. Conclusions

In an unpredictable pandemic situation, the app we developed proved to be reliable and useful in overcoming language barriers by conveying verbal information about SARS-CoV-2 vaccinations to vaccination candidates with no or very limited proficiency in the local language. The main reason for the app’s success was the user- and practice-oriented presentation of the content. The app demonstrated its usability in promoting equitable access to vaccination services and raising the SARS-CoV-2 vaccine willingness among people with limited proficiency in the local language.

## Figures and Tables

**Figure 1 vaccines-10-00360-f001:**
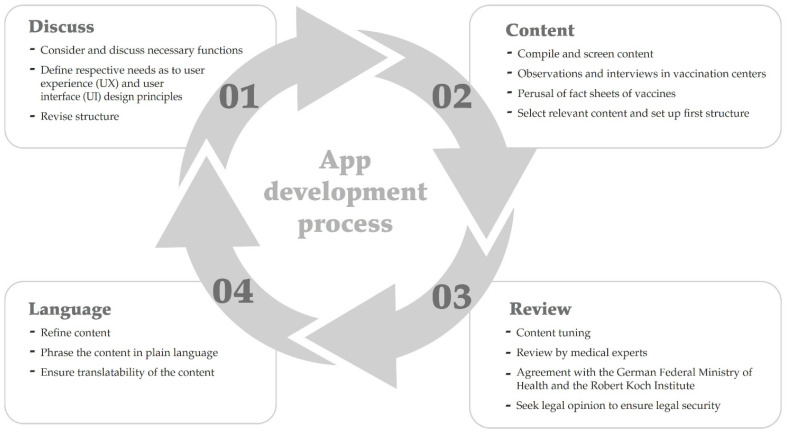
A graphic of the app development process. Cycles were iterative and the tasks within each cycle were performed for each update.

**Figure 2 vaccines-10-00360-f002:**
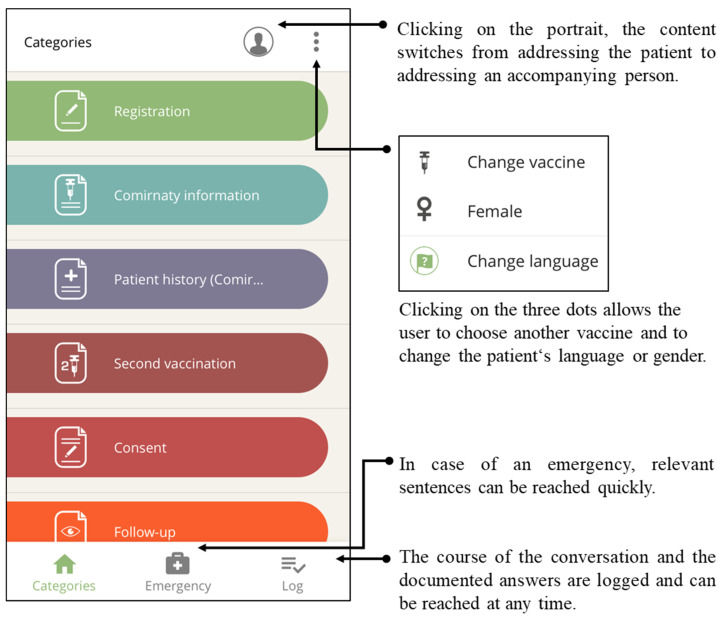
The content is grouped into categories with recognizable icons. The user interfaces allow changes in addressee, administered vaccines, chosen language, and the vaccination candidate’s gender at any time.

**Figure 3 vaccines-10-00360-f003:**
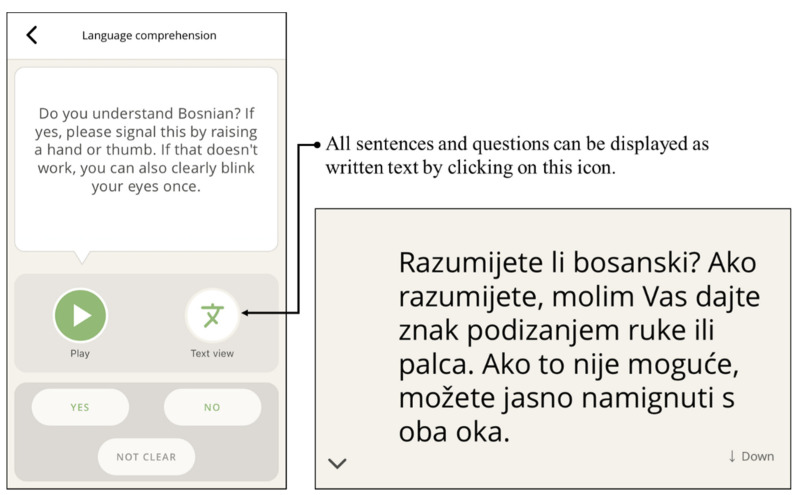
Language comprehension can be checked in the first step (here in Bosnian: “Do you understand Bosnian? If yes, please signal this by raising a hand or thumb. If that doesn’t work, you can also clearly blink your eyes once.”). The phrases can be played back audibly or displayed as text.

**Figure 4 vaccines-10-00360-f004:**
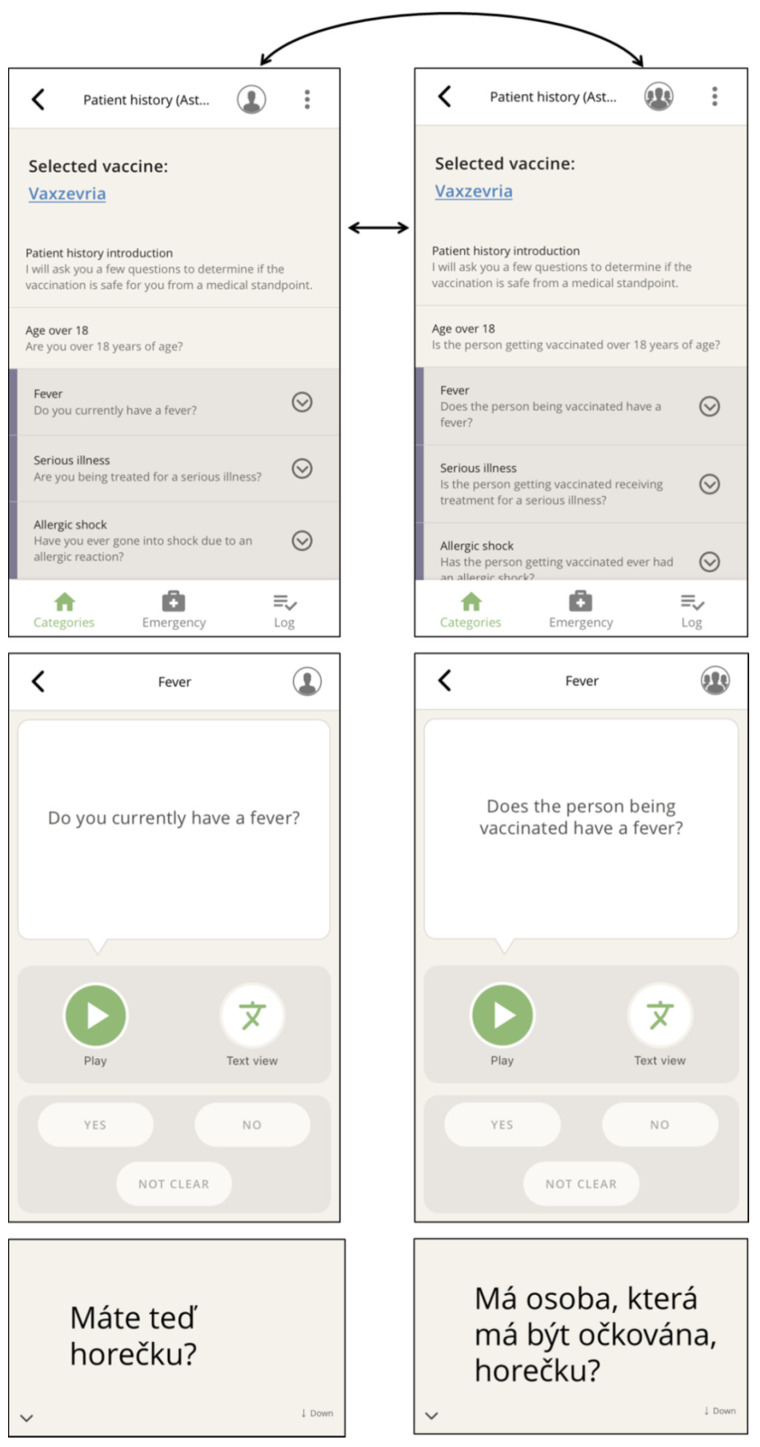
The addressed person can be toggled between by clicking on the portrait icon within categories (**top**) or directly on the selected question (**center**). The respective text display (**bottom**) can be reached via the text view button (smaller-scale screenshot rotated 90°) (here, Czech: “Do you currently have a fever?” and “Does the person being vaccinated have a fever?”).

**Figure 5 vaccines-10-00360-f005:**
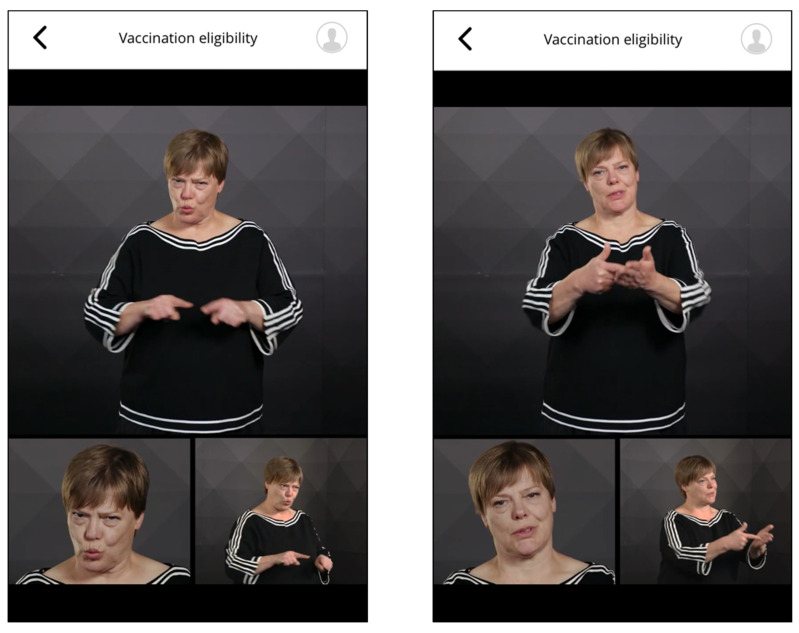
Screenshot of German sign language (“Can you please show me your vaccination invitation?”).

**Figure 6 vaccines-10-00360-f006:**
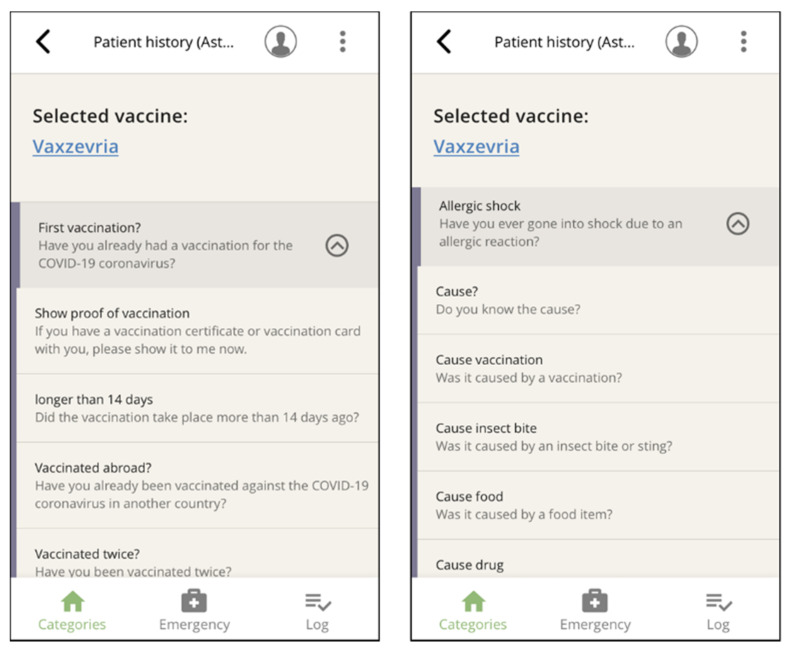
Two examples of filter questions that trigger related questions.

**Table 1 vaccines-10-00360-t001:** Languages and dialects (*n* = 39) supported by the app (in alphabetical order).

Language/Dialect		
Albanian	French	Romanian
Arabic	German (incl. sign language ^1^)	Russian
Bosnian	Greek	Serbian
Bulgarian	Hebrew	Slovakian
Chinese (China)	Hindi	Sorani (Central Kurdish)
Chinese (Taiwan)	Hungarian	Spanish (Spain)
Croatian	Italian	Spanish (Latin America)
Czech	Kurdish-Sorani	Swedish
Danish	Lithuanian	Tamil
Dari (Persian)	Pashto (Afghani)	Thai
Dutch	Polish	Turkish
English	Portuguese (Portugal)	Ukrainian
Farsi (Persian)	Portuguese (Brazil)	Vietnamese

^1^ Any language except German sign language can also be selected as the app’s operating language.

## Data Availability

Data of the pilot test are available from the authors upon request and within a data sharing agreement.

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
