# Peer review of "A Multilingual App for Providing Information to SARS-CoV-2 Vaccination Candidates with Limited Language Proficiency: Development and Pilot"

_vaccines, 2022, doi:10.3390/vaccines10030360_

Round 1
Reviewer 1 Report
Thank you for the opportunity to review this interesting paper. The authors have developed an app that is designed to provide information and guidance throughout the SARS- CoV- 2 vaccination progress in multiple languages and sign language.
The concept is innovative and addresses a relevant gap and population at risk. However, the app has only been tested in a small pilot group so far and it is unclear whether and to what extent the participants profit from the information provided by the app. The evaluation of the app by the participants is decribed in little detail. Therefore it is diffcult to assess if it was better than the standard information especially as there was no control group in place.
More detailed comments:
Abstract:
Results are kept very short and provide little information regarding the outcome
Minor: abbreviation HCP needs to be introduced
Introduction:
There is no information provided whether there are similar apps providing information on SARS- CoV- 2 or other diseases world wide.
Material and methods:
The process of developing the app (STARS steps) have been described in detail. This part would benefit from shortening. On the other hand, the set up at the vaccination centre was decribed only briefly.
How are the participants supposed to use the app? Download it on their phone while the mobile team was there? Or did you provide tablets? How much time did you allow the participants for app use?
Results:
The information provided in the "results" section about the developemnt of the app is very detiled and could be more concise, the data on the pilot project and especially evaluation of results, however, would profit from more information on evaluation by the participants and the HPC (no number, no graded information) when it comes to the benefit of the app. E.g. on page 13, line 359-361 it is unclear how compliance was improved by app use.
Conclusion:
Where do you present the data that proves the advantage of the single phrase categories?
Author Response
Department of General Practice
University Medical Center Göttingen / Georg-August-University
Humboldtallee 38
37073 Göttingen, Germany
To
Ms. Regina Qi
Reviewers
MDPI Vaccines Editorial Office
St. Alban-Anlage 66, 4052 Basel, Switzerland
Manuscript ID: vaccines-1558604 - Major Revisions by 24 January 2022
Dear Ms. Qi, dear Sir or Madam,
Thank you very much for accepting a revised version of our manuscript “A multilingual app for providing information to SARS-CoV-2 vaccination candidates with limited language proficiency: Development and pilot” to be considered for publication in ‘Vaccines’. We would like to express our thanks to the reviewers for their efforts and appreciate the many helpful comments that improved our manuscript considerably.
The reviewers gave conflicting recommendations regarding the sections’ lengths and how detailed the development process needs to be outlined: Reviewer #1 and #3 suggested a more concise description of applied methods and the resulting app while reviewer #2 asked for a more detailed description on how the approach was carried out in each of the project phases. In principle, both points of view are reasonable, but in view of the aims and scopes as well as the anticipated audience of the journal, we preferred to rather shorten the manuscript and keep it concise. However, we would like to leave the final decision to the editor.
We revised the manuscript according to the reviewers’ comments using the “Track Changes” function. In the following, we will comment on our revisions and respond to reviewers’ comments.
Yours sincerely,
Jennifer Schäning, Frank Müller, and Eva Maria Noack
Reviewer 1:
Thank you for the opportunity to review this interesting paper. The authors have developed an app that is designed to provide information and guidance throughout the SARS- CoV- 2 vaccination progress in multiple languages and sign language.
The concept is innovative and addresses a relevant gap and population at risk. However, the app has only been tested in a small pilot group so far and it is unclear whether and to what extent the participants profit from the information provided by the app. The evaluation of the app by the participants is decribed in little detail. Therefore it is diffcult to assess if it was better than the standard information especially as there was no control group in place.
- Thank you very much for your comment. We agree that for a further study it is a very important task to assess efficiency or an improvement of knowledge towards SARS-CoV-2 vaccinations in foreign-language speaking vaccine candidates. Our chosen study design is not sufficient to do so as it was not the study’s objective. In the pilot study, we aimed to assess a general feasibility for using our tool in a real world setting, to dismantle potential hurdles in using it and provide feedback for further development. We added this aspect in the paper.
The currently often-performed “usual care” practice relying solely on multilingual paper forms (if available in the vaccination candidates’ languages) is a neither ethically nor legally compliant way to obtain informed consent. The use of the app can therefore by its very nature be seen as an improvement.
We added both aspects to the ‘limitations and strengths section’ in the discussion:
“The pilot test was aimed at assessing a general feasibility for using our tool in a real-world setting, at identifying potential hurdles of usage and at feedbacking the development process. This pragmatic pilot test with 20 observed uses was, by its very nature, not suitable to assess the efficacy of the app as an intervention. Further research is needed to determine, whether vaccination candidates are better informed through the app than through multilingual information forms. Nevertheless, it must be noted that multilingual forms can never replace a legally and ethical sound informed consent talk. Here the app offers a significant improvement.”
More detailed comments:
Abstract:
Results are kept very short and provide little information regarding the outcome
- The app itself can be considered as main output of the project. As stated above, the pilot test was conducted to get first insights about how the app is used in a real world setting and was meant to feedback further development steps. However, the term “pilot study” that also occurred in the manuscript can be misleading. We changed this term to “pilot test” throughout the manuscript.
Additionally, we rephrased the abstract to clarify the purpose of the pilot test. It now reads:
“Language barriers are obstacles to take vaccinations against COVID-19. They jeopardize informed consent, vaccination safety and a positive immunization experience. We have developed a multilingual app to overcome language barriers when dealing with vaccination candidates with limited proficiency in the locally spoken language. We applied the Spiral Technology Action Research (STAR) model to create the app within a discursive process involving healthcare professionals (HCPs) from vaccination sites, literature searches and guidelines and field trials at vaccination centers. In a real-world pilot test, we assessed usability and feedback for further improvement. Our efforts resulted in an app that facilitates communication with vaccination candidates in 40 languages with each over 500 phrases that can be played back or displayed as text. In the pilot test, the app demonstrated its usability, and was well accepted by the vaccination candidates (n=20). The app was mainly used to inform about risks and benefits of SARS-CoV-2 vaccination. Some HCPs struggled to navigate the comprehensive content and the pilot test exposed the need for additional phrases. The STAR model proved flexible for adapting to dynamic pandemic conditions and changing recommendations. This multilingual app overcomes language barriers in healthcare settings promoting vaccines to migrants with limited language proficiency.”
Minor: abbreviation HCP needs to be introduced
- Thank you for this remark. We changed it accordingly to your suggestion.
Introduction:
There is no information provided whether there are similar apps providing information on SARS- CoV- 2 or other diseases world wide.
- In German vaccination centers, multilingual forms can be available for foreign-language vaccination candidates (we haven’t checked if they are always available). The WHO launched multilingual leaflets and some European governments provide also multilingual information videos that can be used for information purpose.
To the best of our knowledge, there is no app that supports HCP to overcome language barriers for informed consent talks on SARS-CoV2-vaccination or any other vaccinations. We state so in the discussion section.
Material and methods:
The process of developing the app (STARS steps) have been described in detail. This part would benefit from shortening. On the other hand, the set up at the vaccination centre was decribed only briefly.
- The development process is indeed a bit wordy. We have shortened it and added some more information about the regular procedures in the vaccination centers.
How are the participants supposed to use the app? Download it on their phone while the mobile team was there? Or did you provide tablets? How much time did you allow the participants for app use?
- As the app is solely administered by HCPs, thus vaccination candidates do not need to bring any digital device with them. We have clarified this in the manuscript:
“The app is designed to be operated by HCPs on their (work) smartphones or tablets. Thus, vaccination candidates neither need to install an app nor have a smartphone at all.”
The staff of the mobile teams we surveyed could choose to install and use the app on their (work) smartphones or use one of the tablets that we provided. We added this aspect:
“HCPs could use the app on their own smartphones or on one of the tablets we provided (Apple iPad mini 4).”
As we aimed to get an unbiased impression of the actual usage, we did not apply any guidelines or restrictions how HCPs should use the app to communicate with their vaccination candidates. We added these aspects to the methods section:
“In order to get as unbiased an impression as possible of how the app is actually used, HCPs did not receive any instructions beyond a short introduction showing the basic functions. Especially, we did not specify how the app should be used for consent talks and left this to the decision of the HCPs. We did not set any time restrictions for the use of the app. The observers did not intervene, help or comment in the app use situation.”
Results:
The information provided in the "results" section about the developemnt of the app is very detiled and could be more concise, the data on the pilot project and especially evaluation of results, however, would profit from more information on evaluation by the participants and the HPC (no number, no graded information) when it comes to the benefit of the app. E.g. on page 13, line 359-361 it is unclear how compliance was improved by app use.
The results are meant to describe in detail the app we have developed as the paper’s focus lies on the results of the development process. Another reviewer had requested a more detailed description of the concrete steps, so we have tried to find a balance here.
We agree that more information about HPCs would be helpful to interpret the results and to learn how individual HPCs take up new technology. However, it was our objective to learn about the overall approach, about how the app is used and whether vaccine candidates complied with this type of digitally supported consent talk. To operationalize this, we used a 5-point Likert scale to indicate how the vaccination candidates interacted and complied with the situation, that information was provided and questions posed to them by an app. Since the previous description of the observational protocol was misleading, we slightly amended the paragraph. It now reads:
“The observation protocol was concluded by three questions on the overall assessment, which also had to be rated on a five-point Likert scale. These questions focused on the overall cooperation/compliance of the vaccination candidate with the situation that information was provided and questions were posed by the app and whether it was possible to communicate with the vaccination candidate in German at all.”
Conclusion:
Where do you present the data that proves the advantage of the single phrase categories?
- Thank you for your remark. Indeed, this aspect was presented unclear. We rephrased the sentence as following:
“The main reason for the app’s success was the user- and practice-oriented presentation of the content.”
Reviewer 2 Report
This paper describes the creation of an app’s content with 40 languages to support the German vaccination program. It applies STAR model with 14 discursive process involving medical experts and staff from vaccination sites. Some of the drawbacks of the paper are: 1. It does not demonstrate clearly in each step how STAR model was applied. 2. The evidence based scientific soundness is not well-established as the outcomes from various groups of users and their differences catered in the app are not clear for a reader. 3. Example scenarios of the app could be shown to demonstrate the use, advantages and efficacy of the app. 4. A section on limitations / constraints and further research should be included. For example, the perspective of migrant groups was not directly collected. However, no mention regarding if future improvements could include this. There are several unanswered questions too. Can the app be used in other countries? How much of generalisation is done? What are the customisations? Without any screenshot example case scenarios, it is diffocult to visiualise the app and its usability.
Author Response
Jennifer Schäning, Frank Müller, and Eva Maria Noack
Department of General Practice
University Medical Center Göttingen / Georg-August-University
Humboldtallee 38
37073 Göttingen, Germany
To
Ms. Regina Qi
Reviewers
MDPI Vaccines Editorial Office
St. Alban-Anlage 66, 4052 Basel, Switzerland
Manuscript ID: vaccines-1558604 - Major Revisions by 24 January 2022
Dear Ms. Qi, dear Sir or Madam,
Thank you very much for accepting a revised version of our manuscript “A multilingual app for providing information to SARS-CoV-2 vaccination candidates with limited language proficiency: Development and pilot” to be considered for publication in ‘Vaccines’. We would like to express our thanks to the reviewers for their efforts and appreciate the many helpful comments that improved our manuscript considerably.
The reviewers gave conflicting recommendations regarding the sections’ lengths and how detailed the development process needs to be outlined: Reviewer #1 and #3 suggested a more concise description of applied methods and the resulting app while reviewer #2 asked for a more detailed description on how the approach was carried out in each of the project phases. In principle, both points of view are reasonable, but in view of the aims and scopes as well as the anticipated audience of the journal, we preferred to rather shorten the manuscript and keep it concise. However, we would like to leave the final decision to the editor.
We revised the manuscript according to the reviewers’ comments using the “Track Changes” function. In the following, we will comment on our revisions and respond to reviewers’ comments.
Yours sincerely,
Jennifer Schäning, Frank Müller, and Eva Maria Noack
Reviewer 2:
This paper describes the creation of an app’s content with 40 languages to support the German vaccination program. It applies STAR model with 14 discursive process involving medical experts and staff from vaccination sites.
Some of the drawbacks of the paper are:
- It does not demonstrate clearly in each step how STAR model was applied.
- Thank you for this comment that we have discussed among all authors. Two further reviewers requested to shorten the sections about how we applied STAR model for the development process. We have tried to be more concise at these particular sections without abstaining to report on certain important details. A detailed description of the STAR process with individual steps would certainly be interesting for specialist readers from the field of digital health intervention development, but the article would possibly no longer be within the scopes of the journal. However, we leave a final decision to the editors.
- The evidence based scientific soundness is not well-established as the outcomes from various groups of users and their differences catered in the app are not clear for a reader.
- Indeed, we have structured the input from potential users into broad groups. For example, we did not differentiate between physicians, nurses, paramedics, and administrative staff of vaccination centers and how each contributed in the STAR model but merged them to HCPs.
Our aim was here to contribute to a concise description rather than jeopardizing scientific soundness. In principle, we can break this down in more detail, but this would contradict the recommendations of the other two reviewers for a shorter article. However, we leave a final decision to the editors.
Example scenarios of the app could be shown to demonstrate the use, advantages and efficacy of the app.
- Thank you for this remark. With the manuscript, we provided a screencast, i.e. a digital recording of a screen output, showing one short example to demonstrate the use of the app. This screencast is supposed to be published with the manuscript. As it seems that this has not been provided for you during the review process, we uploaded it here:
https://owncloud.gwdg.de/index.php/s/mxASd6IjpAAY4WL
We hope that this demonstration will contribute to a better understanding of the functioning and potential effect of the app than written example scenarios could do. Still, we added some overall aspects about general advantages of our approach:
“Nevertheless, it must be noted that multilingual forms can never replace a legally and ethical sound informed consent talk. Here the app offers a significant improvement.”
or to highlight that the app enables equitable access:
“Vaccination candidates neither need to install an app nor have a smartphone at all.”
A section on limitations / constraints and further research should be included. For example, the perspective of migrant groups was not directly collected. However, no mention regarding if future improvements could include this.
- We agree, the perspective of foreign-language patient is very important and often neglected. In the limitation and strengths section, we state: “[…] future research should focus on listening to healthcare recipients, i.e. find out how foreign-speaking patients and minority groups receive and interpret medical information.” This applies also for our app, which as to be assessed by the recipients of the information. We observed the app usage, but by an observation we were of course not able to investigate how the app is perceived by the vaccination candidates. We describe this in the limitations and strengths section and conclude “These aspects are thus a good starting point for further research.”
We additionally added that an efficacy study is urgently needed:
“The pilot test aimed at assessing a general feasibility for using our tool in a real-world setting, at identifying potential hurdles of usage and at serving as feedback for the development process. This pragmatic pilot test with 20 observed uses was, by its very nature, not suitable to assess the efficacy of the app as an intervention. Further research is needed to determine, whether vaccination candidates are better informed through the app than through multilingual information forms.”
There are several unanswered questions too. Can the app be used in other countries? How much of generalisation is done? What are the customisations? Without any screenshot example case scenarios, it is diffocult to visiualise the app and its usability.
- The app can be principally used worldwide. It can be downloaded free of charge in all major app stores and the user language can be any of the supported languages. However, the app’s content is tailored to the vaccines approved in Germany and to the recommendations of the German Standing Committee on Vaccination. Other country specific customization has not been covered by the funding as the app meant to support mainly the German vaccination campaign.
We added this important aspect to the outlook section:
“In principle, the use of the app is not limited to Germany, as any supported language can be used as both operating language and output language. The approach could thus also support many international vaccination campaigns. At present, however, only the recommendations of the German Standing Commission on Vaccination are consistently supported.”
Reviewer 3 Report
The authors have done great work by building an app that helps non-native german speakers to communicate with health workers or get vaccine jabs in their language.
Overall, the paper is well written. However, it needs some proofreading to fix some grammatical mistakes. Besides, the article is lengthy and should be shortened.
I wonder why authors have only got 20 vaccination candidates. What is the current user number of app.
Author Response
Jennifer Schäning, Frank Müller, and Eva Maria Noack
Department of General Practice
University Medical Center Göttingen / Georg-August-University
Humboldtallee 38
37073 Göttingen, Germany
To
Ms. Regina Qi
Reviewers
MDPI Vaccines Editorial Office
St. Alban-Anlage 66, 4052 Basel, Switzerland
Manuscript ID: vaccines-1558604 - Major Revisions by 24 January 2022
Dear Ms. Qi, dear Sir or Madam,
Thank you very much for accepting a revised version of our manuscript “A multilingual app for providing information to SARS-CoV-2 vaccination candidates with limited language proficiency: Development and pilot” to be considered for publication in ‘Vaccines’. We would like to express our thanks to the reviewers for their efforts and appreciate the many helpful comments that improved our manuscript considerably.
The reviewers gave conflicting recommendations regarding the sections’ lengths and how detailed the development process needs to be outlined: Reviewer #1 and #3 suggested a more concise description of applied methods and the resulting app while reviewer #2 asked for a more detailed description on how the approach was carried out in each of the project phases. In principle, both points of view are reasonable, but in view of the aims and scopes as well as the anticipated audience of the journal, we preferred to rather shorten the manuscript and keep it concise. However, we would like to leave the final decision to the editor.
We revised the manuscript according to the reviewers’ comments using the “Track Changes” function. In the following, we will comment on our revisions and respond to reviewers’ comments.
Yours sincerely,
Jennifer Schäning, Frank Müller, and Eva Maria Noack
Comments of Reviewer 3:
The authors have done great work by building an app that helps non-native german speakers to communicate with health workers or get vaccine jabs in their language.
Overall, the paper is well written. However, it needs some proofreading to fix some grammatical mistakes. Besides, the article is lengthy and should be shortened.
I wonder why authors have only got 20 vaccination candidates. What is the current user number of app.
Answer to Reviewer 3:
- Thank you very much for your time in reviewing our manuscript. Especially in development projects where much effort is spent in small details that are often not immediately seen by others but makes the difference in the end, it is good to know that others acknowledge this work.
We have proofread the manuscript and corrected some typos.
We shortened parts of the manuscript, also following the suggestions of the other reviewers.
The app is currently installed on over 73,000 devices. The vaccination mode was integrated into an existing app designed for paramedics to overcome language barriers. However, as many paramedics are also providing their workforce to the vaccination campaign, it was deemed beneficial to include it in this existing and widespread app.
As due to privacy protection regulations we cannot track which of the app’s content is used, we cannot comment reliably on how many informed consent talks have been supported by the app. However, the high distribution as well as feedback from users as well as the experience in this study during the pilot testing where all HCPs already were aware of the app, some had installed it, and some were already using it regularly and frequently suggests that it is currently used manifold times.
We agree that our pilot test was rather pragmatically carried out. At first, we aimed to observe the app use in German vaccination centers. However, very few if any foreign-language speaking vaccination candidate were reaching out to these facilities, as scheduling for an appointment required extensive use of German language (e.g. speaking with a German call center agent, scheduling an appointment on a German website, proofing eligibility with documents, reading of leaflets and filling out forms in German language). Eventually, we surveyed ‘Mobile teams’ that reach out to all those that cannot reach mass vaccination sites, e.g. in nursing care, homeless shelters or among underserved communities. As a walk-in concept was applied here, we could not plan in advance if vaccine candidates with limited German language proficiency would show up. Additionally, a pilot test was not foreseen by the funding body so we decided to leave it at 6 outreaches.
We agree that an extended study of the app's effectiveness is urgently needed. In this paper, we describe a necessary step along the way and invite colleagues to benefit from our knowledge and experience as they embark on a similar effort.